# Factors Associated with Dual-Fluency Walk Speed Improvement after Rehabilitation in Older Patients

**DOI:** 10.3390/jcm11247443

**Published:** 2022-12-15

**Authors:** Marion Cubillé, Catherine Couturier, Guy Rincé, Thibault Deschamps, Pascal Derkinderen, Laure de Decker, Gilles Berrut, Guillaume Chapelet

**Affiliations:** 1Clinical Gerontology Department, Centre Hospitalier Universitaire (CHU) de Nantes, 44000 Nantes, France; 2Movement, Interactions, Performance, MIP, CHU de Nantes, Nantes Université, EA 4334, 25 Bis Boulevard Guy Mollet, BP 72206, 44322 Nantes, France; 3Neurology Department, CHU de Nantes, 44000 Nantes, France

**Keywords:** walk speed, older adults, rehabilitation, dual-task

## Abstract

Walk speed measured under dual-task conditions (neurocognitive tasks) could reflect patient performance in real-life. Rehabilitation programs are effective in increasing walk speed, but few studies have evaluated the associations between geriatric factors and rehabilitation efficacy under dual-task conditions. Our objective was to investigate the association between geriatric factors and an increase in dual-task walk speed (threshold of 0.1 m/s), after a multidisciplinary rehabilitation program. We performed a retrospective cohort study that included patients aged 75 years and over, who underwent a complete rehabilitation program and who had a neurocognitive assessment at baseline. The primary outcome was the increase in the dual-task (fluency verbal task) walking speed between pre- and post-rehabilitation assessments. In this study, 145 patients were included, with a mean age of 83.6 years old. After rehabilitation, dual-task walk speed increase in 62 (43%) patients. In multivariate analysis, the following factors were associated with an increase in dual-task walk speed: IADL (OR 2.50, 95% CI [1.26; 4.94], *p* = 0.009), vitamin D level (OR 0.83, 95% CI [0.72; 0.95], *p* = 0.008), severe sarcopenia (OR 0.00, 95% CI [0.00; 0.32], *p* = 0.016), depression (OR 15.85, 95% CI [1.32; 190.40], *p* = 0.029), number of drugs (OR 1.41, 95% CI [1.04; 1.92], *p* = 0.027), initial dual-fluency walk speed (OR 0.92, 95% CI [0.86; 0.98], *p* = 0.014) and time interval between initial and final assessments (OR 0.98, 95% CI [0.96; 1.00], *p* = 0.06). Identifying patients that are less resilient to rehabilitation may promote a centered-patient approach for an individualized and optimized rehabilitation care.

## 1. Introduction

Walk speed, often called the sixth vital sign, could be considered as a clinically relevant health biomarker [1]. Indeed, walk speed predicts the subsequent course of many outcomes, including mobility, cognitive decline, incidence of falls and fractures, independence in activities of daily living, general health status, recovery from an acute medical episode, frequency of hospitalization, institutionalization and overall mortality [2]. Recently, world guidelines highlighted that gait speed is one of the three main questions for risk assessment and management for community-dwelling older adults [3].

Assessing walk speed under dual-task conditions (combined with a cognitive task, such as verbal fluency task) could approach locomotion abilities under real-life conditions [4]. In addition, walk speed assessment, under dual-task condition, could increase the sensitivity of detection of health outcomes, such as fall risk and walk disorders [4]. The first to conceptualize and understand this were Lundin-Olsson et al. who created the “stop talking when walking test” [5]. Later, several studies have shown that an impairment in dual-task walk speed was associated with an increased risk of fall and a decline in cognitive function [6,7,8,9,10,11].

Several studies have shown that it was possible to create rehabilitation programs to increase dual-task walk speed in older patients. [12]. However, in the literature, these studies present contradictory results about the main factors that are associated with rehabilitation effectiveness [13,14,15,16]. For example, patients with cognitive disorders were associated with rehabilitation effectiveness in some studies [13,14], but other intervention trials showed only limited effectiveness of physical training in patients with cognitive impairment [16]. It could be possible that these conflicting results were related to the heterogeneity of the population and to the methodology of the cognitive assessment methods. Indeed, the different domains of cognition are unequally involved in locomotor control and dual-task assessment. For example, executive cognitive functions, which are the most commonly associated with the alteration of walk parameters, are not systematically evaluated in rehabilitation programs [11]. Finally, in older patients the search for an association between an increase in dual-task walk speed after rehabilitation and cognitive functions may be biased by confounding factors such as depression, iatrogeny, osteoarticular pain and malnutrition [3]. Considering this, further research is needed to explore the association between geriatric assessment and dual-task walk speed in rehabilitation programs. Moreover, identifying the factors associated with rehabilitation efficacy could help to promote effective, individualized and optimized rehabilitation programs in the future [17]. Indeed, identifying patients that are less resilient to rehabilitation may allow for better referral of these patients in the care system and promote a centered-patient approach for an individualized and optimized rehabilitation care.

Considering this, the aim of this study was to explore the association between an increase in dual-task walk speed after rehabilitation and the geriatric factors, identified by a geriatric assessment.

## 2. Materials and Methods

### 2.1. Study Population

This retrospective study was conducted in a tertiary, 2600-bed, university-affiliated center, at the Clinical Gerontology Department, Nantes, France. In this department, after a geriatric assessment, a multidisciplinary care program is proposed to older patients who have fallen [18]. The multidisciplinary care program included the participation of geriatricians, physiotherapists and nurses who provided patients with heath advice, such as nutritional, physical and social advice. Moreover, patients could be referred to a psychologist if necessary.

From October 2010 to March 2019, all patients who were 75 years of age or older were retrospectively included if they met all the following criteria: (i) participation rehabilitation program, (ii) presence of a neurocognitive assessment (MMSE over 18 and Frontal Assessment Battery assessment). For each patient, only the first rehabilitation program participation was considered and there were no other exclusion criteria.

### 2.2. Walk Assessment and Rehabilitation Program

All patients were first seen in consultation by a geriatrician who performed a geriatric assessment. Following this consultation, the rehabilitation program was proposed to all patients who agreed to come to the Gerontology Department twice a week [18]. Patients who agreed to participate to the rehabilitation program received an initial and final walk speed assessment by a physiotherapist. In detail, the rehabilitation program consisted of individual sessions of 45 min, twice a week, for 7 weeks. This rehabilitation was personalized and progressive, adapted to the deficits identified during the initial assessment and evolved according to the progress generated. It aimed to improve balance by reducing instability, increase walking speed, improve the sit-to-stand transfer, make movement the patient safer, assist in lifting from the ground, strengthen physical performance and reduce falls and the fear of falling. Each session included a time of active relaxation, muscle strengthening and overall balance rehabilitation. Techniques consisted of reducing the support surfaces, working on supports inducing sensory disturbances to develop postural adaptation reactions, rehabilitation of parachute reactions and reduction in oscillations. Finally, walk rehabilitation aimed to increase walking speed by working on stride length, cadence, anticipation of changes in direction and endurance. Adapted walking courses were proposed indoors and outdoors, with variable obstacles and with the introduction of dual attentional tasks.

At the beginning and at the end of the rehabilitation program, walk speed was measured with the GAITRite^®^ type mat (Biometrics, Gometz-Le-Chatel, France), which quantifies the spatio-temporal data of the dual-task walk. For dual-fluency walk speed assessment, the patient was asked to walk at normal walk speed, while naming as many animals as possible. The assessment was completed with stabilometry, balance and muscle performance tests. The grip strength was evaluated with a Jamar^®^ dynamometer (Performance Health, Reims, France) and consisted in measuring the dominant hand grip (arm alongside the body and the elbow bent at 90°).

### 2.3. Data Collection

The primary endpoint of the study was the effectiveness of the rehabilitation, defined by a gain of at least 0.1 m/s in the dual-fluency walk speed, between pre and post-rehabilitation. We considered this clinically meaningful change of 0.1 m/s because it was associated with a significant gain of 1 point on the instrumental autonomy scale (IADL scale, Instrumental Activities of Daily Living) at 6 months [2] and it is concordant with outcomes assessment, such as fall risk [3].

The primary outcome was retrospectively collected from the hospital electronic medical record. The following data were also collected because of their association with walk speed and/or fall risk: age, gender [19], history of fall, Charlson comorbidity score [20], number of drugs and psychotropic drugs, Body Mass Index (BMI), plasma vitamin D, severe sarcopenia (defined by the combination of a Timed Up ang Go (TUG) greater than or equal to 20 s and a Hand-grip of less than 27 kg for men and 16 kg for women, according to the European recommendation on sarcopenia [21]), MMSE score [22], FAB score [23], time interval between initial and final assessments, ADL (Activities of Daily Living) and IADL autonomy scales [24,25], presence of a depressive syndrome, sensory disorders of the lower limbs, osteoarthritis, and reduced visual acuity reported by the geriatrician during the pre-rehabilitation consultation.

### 2.4. Statistical Analysis

Patients’ characteristics were described using means and standard deviations for continuous variables and frequencies and percentages for binary variables. The variable of interest was the change in dual-fluency walk speed between initial and final evaluations, a binary variable defined as follows: gain less than 0.1 m/s on the one hand, variation greater than or equal to 0.1 m/s on the other hand. The comparison between the two groups was made with the Chi-square test for the binary variables. The normality of distribution of continuous variables was determined by the Shapiro-Wilk test. The variables of non-normal distribution were compared by the Mann-Whitney test, the variables of normal distribution by the student’s *t*-test. Univariate and multivariate logistic regression analyzes were performed to determine associations between variation in dual-fluency walk speed and other variables. The variables included in the logistic regression model for multivariate analysis are those variables that were significantly associated with the variable of interest in univariate analysis and/or with *p* < 0.2, or that are clinically relevant. In addition, strongly correlated variables (Pearson’s correlation coefficient *p* > 0.7) were not included in the model. Relative risks are expressed as odds ratios with 95% confidence intervals. All *p*-values < 0.05 are considered statistically significant. Statistical analyzes were performed on SPSS software version 15.0 (SPSS, Inc., Chicago, IL, USA).

Because of the retrospective cohort study design, no sampling size calculation was performed and all the patients included were considered in the statistical analysis.

### 2.5. Ethical Considerations

The study was conducted in accordance with the ethical principles of the 1983 Declaration of Helsinki and received a positive approval from the Groupe Nantais d’Ethique dans le Domaine de la Santé. Patient identification information was recorded in a secure electronic database.

## 3. Results

### 3.1. Patient Characteristics

We identified 162 patients (Figure 1) who participated in the complete rehabilitation program (7 weeks), between October 2010 and March 2019. Among them, there were four duplicates (i.e., four patients who participated twice to the rehabilitation program), five patients did not have FAB realization and eight patients for whom the initial and/or the final fluency walking speed was missing. Finally, 145 patients were included in the study. The characteristics of the patients are detailed in Table 1.

### 3.2. Factors Associated with an Increase in Dual-Task Walk Speed

Walk speed increased by at least 0.1 m/s from baseline in 62 (43%) in 145 patients. The characteristics of the patients with an increase in walk speed (*n* = 62) and those without (*n* = 83) are described in Table 1. Comparison between the two groups showed significant differences between the two groups: the MMSE score, which was higher in responders (*p* = 0.028), the plasma vitamin D level at baseline, which was lower in responders (*p* = 0.015), the initial dual-fluency walk speed, which was slower in responders (*p* = 0.001) and the time interval between initial and final assessments (i.e., the time between the first assessment and the end of rehabilitation), which was shorter in responders (*p* = 0.001). No significant differences were found for the other parameters.

The results of the univariate and multivariate logistic regression analysis are presented in Table 2. In univariate analysis, the following three factors were significantly associated with the increase in double-task walking speed: vitamin D level (OR 0.95, 95% CI [0.91; 0.99], *p* = 0.021), initial dual-fluency walk speed (OR 1.03, 95% CI [1.01; 1.05], *p* = 0.000) and the time interval between initial and final assessments (OR 0.99, 95% CI [0.99; 1.00], *p* = 0.015).

In multivariate analysis, the seven following factors were significantly associated with an increase in dual-task (fluency verbal task) walk speed after a complete rehabilitation program: instrumental autonomy, assessed by the IADL scale (OR 2.50, 95% CI [1.26; 4.94], *p* = 0.009), initial blood vitamin D level (OR 0.83, 95% CI [0.72; 0.95], *p* = 0.008), severe sarcopenia (OR 0.00, 95% CI [0.00; 0.32], *p* = 0.016), depression (OR 15.85, 95% CI [1.32; 190.40], *p* = 0.029), number of drugs (OR 1.41, 95% CI [1.04; 1.92], *p* = 0.027), initial dual-fluency walk speed (OR 0.92, 95% CI [0.86; 0.98], *p* = 0.014) and time interval between initial and final assessments (OR 0.98, 95% CI [0.96; 1.00], *p* = 0.06).

## 4. Discussion

Assessing walk speed under dual-task conditions (combined with a cognitive task) could approach locomotor abilities under real-life conditions [4]. Moreover, identifying the factors associated with rehabilitation efficacy could help to promote individualized and optimized rehabilitation programs. To our knowledge, this is the first study that highlighted the factors associated with an improvement in dual-fluency walk speed, after an individual personalized rehabilitation program, in older people, i.e., an increase in walk speed of at least 0.1 m/s [2].

In this study, less than half of the patients had an increase in dual-task walk speed after rehabilitation. This proportion seems lower than that observed in the literature. In particular, Uemura et al. found functional improvement in two thirds of patients receiving identical rehabilitation management in a population of 179 elderly patients [13]. However, their primary endpoint was not the same as in our study (TUG test versus dual-task walk speed). We chose our primary endpoint because of its clinical relevance, given its association with improvement in IADL at 6 months [2]. We speculate that these different proportions of improvement following rehabilitation may be related to the use of different endpoints. Indeed, Plummer et al. [12], in a meta-analysis of the effect of rehabilitation on functional abilities in the dual-task condition in older patients, found a mean gain in walk speed in the dual-task verbal fluency condition of 0.09 m/s, which is close to the threshold of 0.1 m/s that we chose for our study.

We observed a better rehabilitation efficacy in patients with a higher IADL score at inclusion. We assume that this result could be explained by the presence of a greater physiological reserve in patients with a high IADL score at baseline, that could potentiate the impact of rehabilitation. Indeed, the IADL score is a scale strongly associated with the evolution of morbidity and mortality [26]. Thus, patients with preserved instrumental autonomy would present a healthier ageing and could be more likely to benefit from rehabilitation. Furthermore, in a study of 432 older patients, Weiss et al. found a significant association between the IADL score and the TUG test [26]. Moreover, autonomy for instrumental daily activities could also reflect the subject’s executive functioning [27]. This result is compatible with the hypothesis that executive dysfunction could have a negative impact on walk improvement after rehabilitation.

Our study showed a greater effectiveness of rehabilitation in patients who were initially vitamin D deficient. This result is counter-intuitive considering the impact of vitamin D deficiency on the central nervous system and the musculoskeletal system [28]. To explain this, we assume that, in this study, the variable ‘vitamin D deficiency’ may be synonymous with ‘correction of vitamin D deficiency’. Indeed, in our multidisciplinary rehabilitation program, presence of vitamin D deficiency systematically led to a medical prescription of vitamin D for correction. In a randomized controlled clinical trial of 148 patients over 60 years of age, Aoki et al. showed that supplementation of vitamin D deficiency resulted in an increase in muscle mass and an improvement in functional ability [28]. Moreover, in a meta-analysis, Annweiler et al. showed an association between plasma vitamin D levels and walk speed [29]. Finally, considering this, it is not surprising that, in this study, patients had an increase walk-speed after correction of vitamin D deficiency.

In this study, patients with a slow initial dual-fluency walk speed benefited more from rehabilitation. We assume that this result can be explained by a combination of several elements. First, we chose a threshold of 0.1 m/s and it is more likely to observe an increase in walk speed in patients whose initial walk speed was slower, as the possibility of improvement is greater. Secondly, in this study, the population analyzed had few comorbidities (low Charlson index) and a relative preserved autonomy (high ADL and IADL scales). The previous variable are known to be associated with successful ageing, less frailty and better resilience [30]. Thus, we assume that in our study the patients with a low walk speed at baseline were resilient and, therefore had a higher probability of benefiting from rehabilitation.

We found that rehabilitation was less effective in patients with severe sarcopenia. This is not surprising because sarcopenia is a global quantitative and/or qualitative alteration of muscles, associated with a decrease in muscle strength [21] and the association between severe sarcopenia and a reduced walk acceleration is intuitive. Indeed, sarcopenia is a muscle disease that impairs physical performance and the ability to respond to rehabilitation.

We found a significant association between the presence of depression and the effectiveness of rehabilitation. It is well known that patients with depression have posture, balance and walk impairment, including a reduced walk speed, and studies have shown the effectiveness of physical exercise in improving motor skills in depressed patients [31,32,33]. This result is therefore consistent with the literature.

In our study, patients who have a greater number of drugs respond better to rehabilitation. This is not surprising because Umegaki et al. showed an association between poly-medication and slower walk speed in elderly adults [34]. This association could be explained by the hypothesis of anticholinergic load, which increases with the number of drugs and which has a negative impact on motor and cognitive abilities [35]. Our study suggests that the negative impact of poly-medication on functional capacities does not hinder the effectiveness of rehabilitation.

We found a significant association between a shorter time between pre- and post-rehabilitation assessments and better rehabilitation effectiveness. This result suggests an association between the frequency of rehabilitation and its effectiveness [18].

Our study found no association between executive syndrome, assessed by the FAB, and the effectiveness of rehabilitation. We believe that this result is mainly related to a too great homogeneity of the executive profile of the studied patients. Indeed, it is well described that not all cognitive and executive functions are equally important in walk control [36]. In our study, we considered the overall FAB score and not its individual items, as they were not detailed in the computerized database. That is why, among the patients included, two patients with an identical score on the FAB could have different executive function impairment. In further studies, it could be interesting to explore the associations between the FAB items and walk speed.

We did not find an association between MMSE and magnitude of improvement in walk speed after rehabilitation. This could be explained by the fact that a small proportion of the population included had neurocognitive disorders (high MMSE and high ADL and IADL). Indeed, in a prospective cohort study, Friedman et al. [22] did show a significant association between MMSE score and walk acceleration after rehabilitation. Uemura et al. [13] also found an association between MMSE score and functional improvement after rehabilitation, as assessed by the TUG, in a cohort of 44 elderly patients with a mean MMSE of 26/30. Several hypotheses can be advanced to explain this association. First, cognitive impairment could be caused by a neurological disorder that also leads to walk disorders. In this case, the impairment could be less reversible after rehabilitation because rehabilitation plays a part only in psychomotor maladjustment, loss of muscle strength and other modifiable factors [36,37]. Second, patients with mild cognitive impairment have a higher rate of their walk impairment related to neurodegeneration and not to others modifiable factors.

This study has limitation. First, it lacks power due to the small number of patients and their heterogeneity, which may alter our conclusions by calling into question statistically significant associations. Second, this was a monocentric study, with a retrospective design. Because of this, we were not able to account for the possible existence of potential confounding factors, such as anticholinergic drugs impact, fear of fall, or stabilometry and balance control analysis [38]. In addition, the single-center design could lead to selection bias and statistical inference bias. Finally, our results should be confirmed in a larger population and with a prospective design.

## 5. Conclusions

Rehabilitation may allow an improvement in the dual-fluency walk speed in older adults. Identifying patient profiles that respond to rehabilitation may allow for better referral of patients in the geriatric care system and adaptation to the increasing demand for care. Further studies are called for exploring more precisely the links between functional improvement and different rehabilitation interventions.

## Figures and Tables

**Figure 1 jcm-11-07443-f001:**
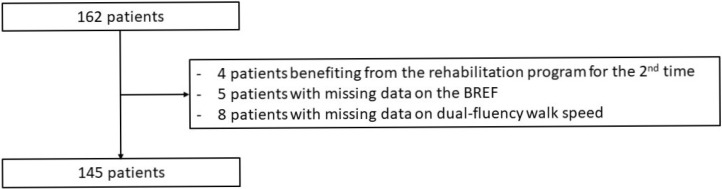
Flow chart.

**Table 1 jcm-11-07443-t001:** Characteristics of the total population and of the subgroups.

	Total Population *n* = 145	Increase in Dual-Task Walk Speed *n* = 62 (43%)	No Increase in Dual-Task Walk Speed *n* = 83 (57%)	*p*-Value
Age in years, mean (SD)	83.6 (5.4)	84.2 (5.4)	83.1 (5.3)	0.228
Female, *n* (%)	103 (71)	47 (76)	56 (67)	0.355
MMSE, mean (SD)	25.6 (3.8)	26.2 (4.0)	25.1 (3.6)	0.028
FAB, mean (SD)	12.2 (4.3)	12.4 (4.2)	12.1 (4.4)	0.678
ADL, mean (SD)	5.4 (1.0)	5.3 (1.0)	5.4 (0.9)	0.773
IADL, mean (SD)	5.5 (2.3)	5.7 (2.3)	5.3 (2.3)	0.292
Charlson Comorbidity Index, mean (SD)	1.1 (1.2)	1.1 (1.3)	1.1 (1.2)	0.997
Number of drugs, mean (SD)	5.7 (3.1)	6.1 (3.1)	5.4 (3.2)	0.138
Number of psychotropic drugs, mean (SD)	0.7 (0.8)	0.8 (0.9)	0.7 (0.8)	0.428
BMI in kg/m^2^, mean (SD)	25.2 (4.8)	25.4 (4.6)	25.1 (4.9)	0.318
Vitamin D (ng/mL), mean (SD)	23.0 (12.0)	19.6 (9.8)	26.1 (13.2)	0.015
Severe sarcopenia, *n* (%)	7 (5)	2 (3)	5 (6)	0.462
History of fall, *n* (%)	117 (81)	48 (77)	69 (83)	0.403
History of repeated falls, *n* (%)	84 (58)	34 (55)	50 (60)	0.610
Osteoarthritis, *n* (%)	46 (32)	18 (29)	28 (34)	0.588
Reduced visual acuity, *n* (%)	58 (40)	26 (42)	32 (39)	0.661
Neurosensorial disorders, *n* (%)	35 (24)	12 (19)	23 (28)	0.243
Depressive syndrome, *n* (%)	54 (37)	27 (44)	27 (33)	0.288
Initial fluency walking speed in m/s, mean (SD)	0.60 (0.20)	0.54 (0.20)	0.65 (0.19)	0.001
Time interval between initial and final assessments, mean (SD)	132 (66)	116 (66)	144 (64)	0.001

SD: Standard Deviation. MMSE: Mini Mental State Examination, FAB: Frontal Assessment Battery, ADL: Activities of Daily Living, IADL: Instrumental Activities of Daily Living, BMI: Body Mass Index.

**Table 2 jcm-11-07443-t002:** Univariate and multivariate logistic regression analyses of factors associated with acceleration of dual-fluency walk speed after rehabilitation.

	Univariate Analysis		Multivariate Analysis	
	OR (IC 95%)	*p*	OR (IC 95%)	*p*
Age	1.04 (0.98; 1.11)	0.227	1.15 (0.98; 1.35)	0.095
Female	1.51 (0.72; 3.17)	0.275	0.95 (0.08; 11.05)	0.968
MMSE	1.08 (0.99; 1.19)	0.096	0.83 (0.62; 1.13)	0.233
FAB	1.02 (0.95; 1.10)	0.607	0.97 (0.76; 1.24)	0.807
ADL	0.94 (0.66; 1.33)	0.720	0.95 (0.39; 2.31)	0.908
IADL	1.08 (0.93; 1.26)	0.307	2.50 (1.26; 4.94)	0.009
Charlson index	1.04 (0.79; 1.36)	0.780	1.61 (0.69; 3.74)	0.268
Number of medications	1.07 (0.96; 1.19)	0.207	1.41 (1.04; 1.92)	0.027
Number of psychotropic drugs	1.19 (0.79; 1.79)	0.404	-	-
BMI	1.01 (0.95; 1.09)	0.685	1.02 (0.87; 1.20)	0.814
Blood vitamin D level	0.95 (0.91; 0.99)	0.021	0.83 (0.72; 0.95)	0.008
Severe sarcopenia	0.48 (0.09; 2.59)	0.397	0.01 (0.00; 0.32)	0.016
History of fall	0.70 (0.30; 1.59)	0.390	0.33 (0.02; 4.35)	0.396
History of repeated falls	0.80 (0.41; 1.56)	0.515	-	-
Depression	1.5 (0.77; 3.01)	0.230	15.85 (1.32; 190.40)	0.029
Osteoarthritis	0.80 (0.39; 1.6)	0.533	0.30 (0.04; 2.57)	0.271
Initial fluency walking speed	0.97 (0.95; 0.99)	0.001	0.92 (0.86; 0.98)	0.014
Time interval between initial and final assessments	0.99 (0.99; 1.00)	0.015	0.98 (0.96; 1.00)	0.026

MMSE: Mini Mental State Examination, FAB: Frontal Assessment Battery, ADL: Activities of Daily Living, IADL: Instrumental Activities of Daily Living, BMI: Body Mass Index.

## Data Availability

Not applicable.

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
