# Peer review of "Factors Associated with Dual-Fluency Walk Speed Improvement after Rehabilitation in Older Patients"

_jcm, 2022, doi:10.3390/jcm11247443_

Round 1
Reviewer 1 Report
Introduction :
First line – reference no:1 – It is a study conducted on a specific population. It is not possible to write a general statement based on this study .
Reference – 2 – Same as the previous comment
What are geriatric parameters?
The rationale of the study is not well supported by the literature
Methodology:
What were the exclusion criteria
What is the design of this study?
From which electronic medical record was the data collected?
The rehabilitation program given to the participants is not clear. I recommend adding a separate table for this.
Why the sample size was not calculated
How do the authors determine whether the participants increased their gait speed or not?
I did not understand why the stabilometry, balance check, and handgrip strength were evaluated in this study. Where are those results?
Where is the data regarding gait speed?
Data analysis and results:
Which are the variable normally distributed?
Discussion :
Each finding of the study has to be discussed properly. The discussion section is incomplete.
Author Response
Introduction :
First line – reference no:1 – It is a study conducted on a specific population. It is not possible to write a general statement based on this study .
We thank the reviewer for this comment. this general statement aims to situate the problematic of the article and the interest to study the walking speed. However, considering this comment, we have corrected this by “could be considered”. We hope that we have well understood this comment.
Reference – 2 – Same as the previous comment
We thank again the reviewer, as our objective was to study factor associated with walk speed increase after rehabilitation, we have considered that, in the first two sentence of our manuscript, it could be relevant to talk about outcomes associated with walk speed.
What are geriatric parameters?
We thank the reviewer for this comment, as this term was not clear, we have decided to switch by “geriatric assessment”.
The rationale of the study is not well supported by the literature
We thank the reviewer for this interesting comment. This is a crucial point. As it is highlighted by the guest editors of the special issue “Updates on Rehabilitation Care for Geriatric Disease”, in older patients, resilience and frailty can hamper the rehabilitation process. Therefore, the outcomes cannot easily be predicted, even though some factors have been associated with rehabilitation effectiveness. Moreover, with an ageing population, there is a crucial need to optimize the practice and effectiveness of geriatric rehabilitation and make it more sustainable. Our study is focused on factors associated with rehabilitation effectiveness, that are not well described in the literature and as it is mentioned by the guest editors. We hope that we have well understood the reviewer comment and that our point of view is correct.
Methodology:
What were the exclusion criteria.
We apologize for this comment. As it is mentioned in the manuscript, the only “non-inclusion” criteria was patient inclusion, more than once, in the rehabilitation program. There was no other exclusion criteria. We had this point in the manuscript.
What is the design of this study?
We apologize for the lack of precision and we thank the reviewer for this comment. This was a retrospective study and we added this precision in the manuscript.
From which electronic medical record was the data collected?
Once again, we apologize for the lack of precision. Data were collected in the hospital electronic medical record. We thank the reviewer for this comment and please see correction in the manuscript.
The rehabilitation program given to the participants is not clear. I recommend adding a separate table for this.
We thank the reviewer for this comment. In our hospital, and as it is mentioned in the manuscript, the rehabilitation program is proposed twice a week to all patients, but rehabilitation is personalized, progressive, and adapted to the deficit and the patient’s capacities. For this reason, it could be difficult to add a separate table. However, the reviewer is right, and this was also noted by the other reviewer. That’s why we have decided to add some precisions concerning the multidisciplinary care program int the first paragraph of the study population section.
Why the sample size was not calculated
We thank the reviewer for this comment that was also noted by the other reviewer. This study was a retrospective cohort study with an exploratory design and no sampling size calculation was performed. We had this in the Material and Method section.
How do the authors determine whether the participants increased their gait speed or not?
We thank the reviewer for this comment, and we apologize for the lack of precision. Our primary outcomes was “the effectiveness of the rehabilitation, defined by a gain of at least 0.1 m/s in the dual-fluency walk speed, between pre and post-rehabilitation” measured by the GAITRite® type mat (Biometrics, Gometz-Le-Chatel, France). We considered this clinically meaningful change of 0.1 m/s because it was associated with a significant gain of 1 point on the instrumental autonomy scale (IADL scale, Instrumental Activities of Daily Living) at 6 months [2].
I did not understand why the stabilometry, balance check, and handgrip strength were evaluated in this study. Where are those results?
We thank the reviewer for these comments. This is a very good point. Stabilometry and balance check analysis were not considered in this study. Because it could be interesting to considered it, we had this in the limitation section.
Considering the handgrip strength test, this was considered in the severe sarcopenia definition : “severe sarcopenia (defined by the combination of a Timed Up ang Go (TUG) greater than or equal to 20 seconds and a Hand-grip of less than 27 kg for men and 16 kg for women, according to the European recommendation on sarcopenia”
Where is the data regarding gait speed?
We thank the reviewer for this comment. In this study, our objective was to explore the factors associated with dual-task gait speed increase and not gait speed. That’s why gait speed was not considered. However, in our population, gait speed and “dual-task” gait speed are highly correlated (Pearson’s correlation coefficient > 0,7).
Data analysis and results:
Which are the variable normally distributed?
In our population, the only two variable that were normally distribued were “Age” and “dual-task walk speed.
Discussion :
Each finding of the study has to be discussed properly. The discussion section is incomplete.
We thank the reviewer for this comment. The context and the main conclusion is discussed in the first paragraph, the primary outcome choice and the rehabilitation effectiveness is discussed in the 2nd paragraph, and all the 7 factors associated with an increased in dual-task after rehabilitation are discussed in the manuscript, i.e. instrumental autonomy (3rd paragraph), initial blood vitamin D level (4th paragraph), initial dual-fluency walk speed (5th paragraph), severe sarcopenia (6th paragraph), depression (7th paragraph), number of drugs (8th paragraph), and time interval between initial and final assessments (9th paragraph). Moreover, we have decided to discussed some non significant results in the 10th paragraph and the study limitations in the 11th paragraph. We considered that this discussion could not be more complete and structured if we wanted to respect the authors guidelines. We apologize for this but we would be delighted if the reviewer could be more explicit for this comment.

Reviewer 2 Report
Abstract:
- The authors refer “The primary outcome was the increase in the dual-task (fluency verbal task) walking speed between pre- and post-rehabilitation assessments”. Taking into account that walking and gait does not mean the same.
Introduction:
- It is not clear what are the contradictory results between studies. It is necessary to mention these contradictions.
Materials and methods:
- Study population:
The study only included participant with history of falls and risk of fall? If so, have to taking into account that your results it is only applicable to this population.
If in the previous publication (ref. 17) the authors included participants of 70 years-old or older, why in this study the court point for the age was 75-year-old?
Whether the authors included all patients who meet the inclusion criteria the statistical inference could be affected. This situation has to been considering at the moment of analysis and discussion.
The reason to not perform a sample design (sampling size calculation and sampling method) must be mentioned.
- Gait assessment and rehabilitation program:
The authors mentioned that the rehabilitation program is a multidisciplinary care program. However, in the description an in the reference 17 only the participation of geriatrician and physiotherapist were mentioned. The program included the participation of other health care professionals (i.e., nurses, psychologist, occupational therapist, etc.)?
- Data collection:
The evidence to use a court point of 0.1 mt/seg as definition of gain in the dual-task gait speed is weak. First, the metanalysis that support this point (ref. 2) consider as end point the gain in the gait speed, not in the dual-task gait speed. Second, heterogeneity of this analysis (the analysis that included all types of exercise) was high. And third, in the reference mentioned none analysis was performed considering the improved of IADL as outcome. I suggest being carefully at the moment to decide the court point.
Considering the relevance of fear of falling in the population of interest, why it wasn't to taking into account as covariable?
The definition of sever sarcopenia included the measuring of muscle mass through a standardized method (i.e., DXA, BIA, MRI or CT).
- Statistical analysis:
In the final logistic regression model, how you controlled the confusion and the interaction?
If your included all the participant, without used a sampling strategy, you can´t used the p-value as measure of statistical significance, or on the other words, of inference.
Results:
- It is not clear the concept of “Time interval between initial and final assessments”. If exist a difference between the moment of final assessment it could introduce a bias in the study.
- The selection of the variables included in the final model is confused. The authors included all variables, except two, in the final model. I suggest consider a different strategy for the selection of variables (i.e., method backward, forward, or stepwise).
Discussion:
- Have to be adjusted accord to the new analyses.
Conclusion:
- “Further studies exploring more precisely the links between functional improvement and different executive functions are needed.” It is not a conclusion of the study.

Author Response
We thank the Reviewers for thoughtful reviews and the view of our manuscript. Please, find below our point-by-point responses to the comments and queries raised by the Reviewers. Please see corrections in green throughout the revised manuscript.
Abstract:
- the authors refer “The primary outcome was the increase in the dual-task (fluency verbal task) walking speed between pre- and post-rehabilitation assessments”. Taking into account that walking and gait does not mean the same.
The reviewer is right and we apologize for this mystake. In order to be clearer, we have decided to replace all the “gait” with “walk” in all the manuscript.
Introduction:
- it is not clear what are the contradictory results between studies. It is necessary to mention these contradictions.
The reviewer is right, we had a sentence, as an example, severity of neurocognitive disorder could be associated with rehabilitation effectiveness is some studies but physical training show only limited effectiveness in some patients with severe neurocognitive disorders . “For example, patients with cognitive disorders were associated with rehabilitation effectiveness in some studies [12, 13], but other intervention trials showed only limited effectiveness of physical training in patients with cognitive impairment [15].”
Materials and methods:
- Study population:
The study only included participant with history of falls and risk of fall? If so, have to taking into account that your results it is only applicable to this population.
The reviewer is right, we have decided to correct this sentence.
If in the previous publication (ref. 17) the authors included participants of 70 years-old or older, why in this study the court point for the age was 75-year-old?
For this study, we decided to include, arbitrarily, patients over 75 years old, because few patients under 70 years old are followed in our center. Moreover, we wanted to keep the cut-off that is chosen to admit patients in the geriatric departement in our Hospital (i.e. 75 years old or older).
Whether the authors included all patients who meet the inclusion criteria the statistical inference could be affected. This situation has to been considering at the moment of analysis and discussion.
The reviewer is right. We decided to list these biais in the limitation paragraph.
The reason to not perform a sample design (sampling size calculation and sampling method) must be mentioned.
This is a relevant comment. This study was a retrospective cohort study with an exploratory design and no sampling size calculation was performed. We had this in the Material and Method section.
- Gait assessment and rehabilitation program:
The authors mentioned that the rehabilitation program is a multidisciplinary care program. However, in the description an in the reference 17 only the participation of geriatrician and physiotherapist were mentioned. The program included the participation of other health care professionals (i.e., nurses, psychologist, occupational therapist, etc.)?
The reviewer is right, patients were also seen by nurses who provided to them many heath advices, as nutritional, and social advices according to local protocols. Moreover, patients could be referred to psychologist if necessary. These sentences were added in the manuscript in the method section.
- Data collection:
The evidence to use a court point of 0.1 mt/seg as definition of gain in the dual-task gait speed is weak. First, the metanalysis that support this point (ref. 2) consider as end point the gain in the gait speed, not in the dual-task gait speed. Second, heterogeneity of this analysis (the analysis that included all types of exercise) was high. And third, in the reference mentioned none analysis was performed considering the improved of IADL as outcome. I suggest being carefully at the moment to decide the court point.
I agree with this very relevant comment. The
Considering the relevance of fear of falling in the population of interest, why it wasn't to taking into account as covariable?
We thank the reviewer for this good point. Fear of fall is very relevant, but we cannot take into account this in this retrospective study. But we had this point in the limitation section.
The definition of sever sarcopenia included the measuring of muscle mass through a standardized method (i.e., DXA, BIA, MRI or CT).
This is a good also a good point. However, in general practice, these equipment are not widely available and are not presently suitable for clinical application. That’s why we decedid to consider only severe sarcopenia, because there is a high correlation between low physical performance measurement and presence of severe sarcopenia (https://pubmed.ncbi.nlm.nih.gov/30312372/ )
- Statistical analysis:
In the final logistic regression model, how you controlled the confusion and the interaction? If your included all the participant, without used a sampling strategy, you can´t used the p-value as measure of statistical significance, or on the other words, of inference.
We thank the reviewer for this crucial point. Statistical inference is concerned with random (sampling) error, i.e. the fact that even a random sample does not exactly reflect the properties of the population and that there is often considerable sample-to-sample variation when we repeatedly draw random samples from a population. https://www.degruyter.com/document/doi/10.5018/economics-ejournal.ja.2020-7/html).
In this retrospective study, we have decided to include all patients because we have considered that patient selection could introduce biais. Moreover, the variables included in the logistic regression model for multivariate analysis are those variables that were significantly associated with the variable of interest in univariate analysis and/or with p<0.2, or that are clinically relevant (i.e known factor to be associated with walk speed). Moreover, if two variable were highly correlated (Pearson correlation coed >0.7 or < -0.7), only one variable was considerer in the multivariable analysis. However, despite this rigorous methodology, the reviewer is right, our study could be biaised, and that is highlighted in the first limitation “ First, this study lacks power due to the small number of patients and their heterogeneity, which may alter our conclusions by calling into question statistically significant associations”. We hope that this sentence will satisfy the reviewer.
Results:
- It is not clear the concept of “Time interval between initial and final assessments”. If exist a difference between the moment of final assessment it could introduce a bias in the study.
We strongly agree with the reviewer. This concept corresponds to the time between the first assessment (start of rehabilitation) and the end of rehabilitation. Indeed, as it is suggested by the reviewer, there were differences in the time to follow-up and intervention between patients, and this could be associated with rehabilitation effectiveness. This may well introduce a bias, and it is precisely for this reason that we have considered this variable in the analysis. However, we apologize if this concept was not clear, and we have decided to correct the manuscript in the result section.
- The selection of the variables included in the final model is confused. The authors included all variables, except two, in the final model. I suggest consider a different strategy for the selection of variables (i.e., method backward, forward, or stepwise).
The reviewer is right, and we apologize for the lack of precision. As it is proposed, if two variables are highly correlated (Pearson’s correlation coef >0.7 or < -0.7), only one variable could be considered in the multivariable model (to preserve the model independence). That’s why, we have decided to look for correlation before the variable’s selection. For the second point, backward, forward, and stepwise strategy, could be considered as strategies to explore association but could be bias if some confounding factors are not included. As our objective was to consider all factor that are known to be associated with walk (that we could collect), we have decided to include all variables. However, thanks to this comment, we have decided to be clearer and to correct the manuscript (method section).
Discussion:
- Have to be adjusted accord to the new analyses.
We thank the reviewer for this suggestion. However, we considered that this comment has been resolved by the answer to the previous point
Conclusion:
- “Further studies exploring more precisely the links between functional improvement and different executive functions are needed.” It is not a conclusion of the study.
We thank the reviewer for this comment and we have switched this sentence by “Further studies exploring more precisely the links between functional improvement and different rehabilitation interventions”.

Round 2
Reviewer 1 Report
The authors tried to improve the manuscript However, there are some major issues that need to be resolved for further consideration
1. The rationale of the study is not well supported by the literature. The introduction must be improved by adding relevant literature
2. I didn’t understand the reason for not calculating the sample size prior to the study. How the authors can be confident about their results.
3. I have a serious concern about the design of the study. An RCT might have been conducted to answer this research question
Please use the word review option for performing revision.
Author Response
We thank the Reviewers for thoughtful reviews and the view of our manuscript. Please, find below our point-by-point responses to the comments and queries raised by the Reviewers.
The authors tried to improve the manuscript However, there are some major issues that need to be resolved for further consideration
- The rationale of the study is not well supported by the literature. The introduction must be improved by adding relevant literature.
We thank the reviewer for this comment and we apologize for that. In this study, our main goal was to highlight the fact that it could be interesting to consider walk speed in dual-task conditions in older patients, during rehabilitation. The rational of our study is supported by the following points, which have structured our introduction.
First, walk speed is associated with clinical outcomes (1,2). Moreover, the recent World guidelines for falls prevention (https://doi.org/10.1093/ageing/afac205) highlighted the fact that gait speed is one of the 3 key questions for fall risk stratification in community-dwelling older adults. However, considering this relevant comment, we have decided to add the previous reference in the introduction to reinforce the interest of our study, i.e focusing on walk speed (ref 3).
Second, walk speed, under dual-task condition, increased the sensitivity of detecting health outcomes (3-10), and could approach real-life conditions. That’s why we assume that it could be interesting to focusing on dual-task walk speed.
Third, several studies, focusing on dual-task walk speed and rehabilitation, have found different results in factors associated with walk speed resilience after rehabilitation (12-15). Considering this, our study modestly attempts to explore this research gap, as it is written in the introduction. Indeed, “identifying patient that are less resilient to rehabilitation may allow for better referral of these patients in the care system and promote a centered-patient approach for an individualized and optimized rehabilitation care”.
We hope that this answer will satisfy the reviewer.
- I didn’t understand the reason for not calculating the sample size prior to the study. How the authors can be confident about their results.
Once again, we thank the reviewer for this point. This is a crucial point. Few studies have tried to focus on the association between dual-task walk speed resilience and rehabilitation, and considering the cut-off of 0,1 m/s as the main outcome. For this reason, it could be difficult, and risky, to calculate a sample size prior the study, that could introduce a bias. Indeed, to purposely estimate a sample size with the assumption that the estimated effect sizes is large (or unknown) can introduce bias (doi: 10.21315/mjms2018.25.4.12).
Moreover, as it is discussed in the following answer to reviewer’s comment, this was an exploratory study, with a retrospective design, and simple size calcul is risky in this kind of study and with a logistic regression analysis (doi: 10.21315/mjms2018.25.4.12).
Some authors recommends a minimum sample size of 500 to derive statistics that can represent the parameters in the targeted population (doi: 10.21315/mjms2018.25.4.12). The other recommended rules of thumb are EPV of 50 and formula; n = 100 + 50i where i refers to number of independent variables in the final model (doi: 10.21315/mjms2018.25.4.12). However, sample size less than 500 may be sufficient for associations that yield medium to large effect size (doi: 10.21315/mjms2018.25.4.12).
Finally, we accept this comment and the fact that our study could be biased. That’s why we have noted these limitations in the last paragraph of the discussion. We hope that this comment will satisfy the reviewer.
- I have a serious concern about the design of the study. An RCT might have been conducted to answer this research question
We strongly agree with the reviewer. It is well known that, before RCT, it is useful to conduct retrospective studies, in order to collect data to optimize the feasibility and the design of a RCT. This study was designed in this objective, i/e designed as an explorative study, with a retrospective study (without calculating a sample size). However, the reviewer is right. The fact that we didn’t have calculated a sample size and the retrospective design (not a RCT) could alter our conclusions. That’s why we have decided to structure the discussion with a point-by-point analysis of our significant results and we have added two limitations associated with the design in the limitation section (please see the first and the second limitations in the last paragraph of the discussion).

Reviewer 2 Report
Congratulations to the authors.
Author Response
Congratulations to the authors.
Once again, we thank again the Reviewers for thoughtful reviews and the view of our manuscript !
